# Nutraceutical-Based Nanoformulations for Breast and Ovarian Cancer Treatment

**DOI:** 10.3390/ijms231912032

**Published:** 2022-10-10

**Authors:** Simona Serini, Roberta Cassano, Federica Curcio, Sonia Trombino, Gabriella Calviello

**Affiliations:** 1Department of Translational Medicine and Surgery, Section of General Pathology, School of Medicine and Surgery, Università Cattolica del Sacro Cuore, Largo F. Vito, 00168 Rome, Italy; 2Fondazione Policlinico Universitario A. Gemelli IRCCS, Largo F. Vito, 00168 Rome, Italy; 3Department of Pharmacy, Health and Nutritional Sciences, University of Calabria, Arcavacata di Rende, 87036 Cosenza, Italy

**Keywords:** breast cancer, in vitro studies, nanoformulations, omega-3 PUFA, ovarian cancer, preclinical studies

## Abstract

Different strategies have been investigated for a more satisfactory treatment of advanced breast cancer, including the adjuvant use of omega-3 polyunsaturated fatty acids (PUFAs). These nutritional compounds have been shown to possess potent anti-inflammatory and antiangiogenic activities, the capacity to affect transduction pathways/receptors involved in cell growth and to reprogram tumor microenvironment. Omega-3 PUFA-containing nanoformulations designed for drug delivery in breast cancer were shown to potentiate the effects of enclosed drugs, enhance drug delivery to target sites, and minimize drug-induced side effects. We have critically analyzed here the results of the most recent studies investigating the effects of omega-3 PUFA-containing nanoformulations in breast cancer. The anti-neoplastic efficacy of omega-3 PUFAs has also been convincingly demonstrated by using preclinical in vivo models of ovarian cancer. The results obtained are critically analyzed here and seem to provide a sufficient rationale to move to still lacking interventional clinical trials, as well as to evaluate possible advantages of enclosing omega-3 PUFAs to drug-delivery nanosystems for ovarian cancer. Future perspectives in this area are also provided.

## 1. Introduction

Over the past few decades, the potential preventive or therapeutic use of nutraceuticals as anti-inflammatory or antineoplastic agents in different pathological settings has represented an area of great interest [1,2,3,4,5,6,7]. In the past, we explored whether combinations of nutraceuticals, known for their anti-inflammatory and/or antineoplastic activities with conventional or innovative anticancer drugs, could potentially enhance the drug effects and ameliorate the physical conditions of patients [8,9,10,11,12]. Most recently, we have also investigated the possibility of improving the action of nutraceuticals or anticancer drugs through their inclusion in nanomaterials with the potential to protect and, more specifically, deliver them to tissues affected by pathological processes [13,14,15,16,17,18]. The strategy of administrating bioactive drugs and/or nutraceuticals in nanoformulations has recently attracted considerable interest worldwide [19]. This nanotherapeutic approach was initially developed since blood vessel endothelium in solid tumors shows enhanced permeability and can be easily crossed in both directions by free antineoplastic drugs and bioactive nutritional compounds having relatively low molecular weight, thus, not allowing an efficient accumulation of these compounds by tumor tissues. On the other hand, nanostructures with a diameter ranging from 10 to 500 nm go through the highly permeable tumor vascular wall but tend to accumulate in the underlying interstitium. These features of the tumor endothelium are known as the permeability and retention effect (EPR), a phenomenon allowing nanoparticles to target tumor cells effectively and passively [20]. In addition, nanoparticle build-up at tumor level has been also related to the insufficient drainage of the fluid escaped from tumor vessels by the lymphatic vessels [21].

Particularly, in recent years, there has been a remarkable upsurge of interest in the potential application of nanotechnologies for the therapy of breast cancer (BCa), which represents the world’s most prevalent cancer [22]. The high rate of frequency of BCa has favored a large-scale mammography screening prevention practice in the female population and its early detection, which has resulted in a considerable decrease in the rate of mortality for this tumor in the last few decades [23]. Nevertheless, despite the early BCa mortality rate has been reduced by modern multimodal therapy approaches, currently, this cancer remains the second-leading cause of cancer-related death in women worldwide [24], by advanced BCa with metastases being a virtually incurable disease, showing a 2/3-year median overall survival [24]. In order to obtain a satisfactory treatment for advanced BCa, many routes are being pursued, including the adjuvant use of nutraceuticals showing antioxidant, anti-inflammatory or antineoplastic properties [25]. The nutraceutical strategy should not be considered as something archaic in comparison to the currently used innovative approaches of personalized targeted therapies and immunotherapies [12]. In fact, some of these dietary compounds were demonstrated to have powerful antiangiogenic effects, the capacity to specifically affect molecular transduction pathways and receptor activities involved in cell growth, as well as to reprogram tumor microenvironments [15,26]. The most recent and relevant advancement in this field of research is the inclusion of these natural bioactive substances in nanoformulations, either alone or in combination with conventional drugs, to better and more specifically deliver them to the neoplastic lesions.

Most of the recently published reports focused on the possibility of using nutraceutical-based nanoformulations for improving the anti-cancer treatment of BCa, flavonoids (such as curcumin, quercetin, epigallocatechin gallate (EGCG), and resveratrol) [27] or garlic constituents [28] have been included in the nanomaterials under study. Starting from these results, two comprehensive reviews of the literature have been recently published [27,28]. Instead, in the first part of this article, we decided to critically analyze the most recent (2017–2022) literature focusing on dietary omega-3 PUFAs enclosed in newly developed nanomaterials for their possible use in BCa therapy. This represents a specific update for BCa, since, in a previous review of ours [15], we already discussed the results of reports that had been published during the period July 2017–July 2018 and that were focused on omega-3 PUFA-containing formulations for the therapy of cardiovascular diseases and cancers of different origin.

Important properties of omega-3 PUFAs may explain the reasons underlying their frequent inclusion in nanoformulations designed for cancer therapy, and BCa in particular. Among them, there are their powerful and widely demonstrated anti-inflammatory and anti-cancer activities [2,29,30,31]. Their anti-inflammatory property has been often invoked to explain their cancer-preventive effects, as well as their efficacy in inhibiting the deleterious side effects frequently associated with chemotherapy [32,33]. However, plenty of preclinical studies have shown that these FAs may exert other and more direct and specific anticancer activities at molecular levels [30,34,35,36,37]. Moreover, it was also shown that they can potentially increase cancer cell sensitivity to conventional antineoplastic therapy [38]. It has been reported that these FAs are able to modulate the activities of a series of membrane-associated receptors and transporters, as well as protein kinases and phosphatases, transcription factors and factors driving and regulating apoptotic or autophagic processes [31]. There is a unifying hypothesis according to which these pleiotropic activities of omega-3 PUFAs may be related to the deep changes that they are able to induce in the lipid microenvironments of cell membranes (lipid rafts) as they become incorporated into them [31]. Consequently, changes in the activity/expression of molecular factors/signaling pathways located there may take place.

Speaking specifically of BCa, it should be underlined that, besides the vast amount of preclinical studies providing evidence for the antineoplastic efficacy of these FAs and exploring their possible mechanisms of action, there exists a consistent body of literature on the outcomes of observational and interventional human studies that substantiate and encourage the possibility of omega-3 PUFA use in the clinical setting (for a review see: [39]).

Another interesting aspect that makes these FAs ideal components of nanoformulations constructed for a more specific delivery to tumor tissues is that cancer cells grow at an abnormally high rate and need great amounts of FAs for the development of their membranes. In particular, cancer cells require and incorporate high amounts of the essential omega-3 PUFA α-linolenic acid (ALA, 18:3 ω-3), which cannot be synthesized by mammalian cells, as well as its metabolic derivatives, the long-chain omega-3 PUFAs eicosapentaenoic acid (EPA, 20:5 ω-3) and docosahexaenoic acid (DHA, 22:6 ω-3). Moreover, the presence of multiple double bonds in their moieties makes them highly peroxidable. Therefore, their inclusion in nanoformulations may help to protect them from oxidative degradation and increase their bioavailability and specific effects at tumor level.

Moreover, recently, considerable effort has been expended in finding strategies that could improve the EPR-induced selective tumor accumulation of drug-delivering nanoparticles by the highly leaky and permeable tumor blood vessels. In fact, whereas a good EPR effect is found in most solid animal tumors rich in blood flow, EPR appears to be inadequate in solid human tumors, especially in some advanced forms of cancer [20]. This has been related to the high interstitial fluid pressure (IFP) and atypical extracellular matrix of the abnormal tumor environment. Consequently, the tumor blood flow becomes heterogeneous and, besides leading to hypoxia and acidosis, it makes difficult the diffusion and uniform distribution of nanoparticles into the vascular tumor area [21]. It has been suggested that one strategy to increase EPR and make nanodrug delivery easier in human tumors is the downregulation of the VEGF signaling and the inhibition of tumor neo-angiogenesis which may produce beneficial changes to the vasculature and IFP reduction [21]. Therefore, nanoformulations able to inhibit angiogenesis, and thus, reduce IFP and improve the delivery of the nanodrugs themselves to tumor tissues have been produced [40,41]. The observation that omega-3 PUFAs possess powerful antiangiogenic properties [42] further indicates the appropriateness of their inclusion in nanoformulations carrying drugs to tumors.

In the last part of this article, we will focus on ovarian cancer (OvCa) and hypothesize the possible therapeutic use of omega-3 PUFAs against it. OvCa is the most fatal among the gynecological malignancies, and its high-mortality rate and poor prognosis have been related to the fact that most OvCa patients are already in the advanced stages of disease when they receive the initial diagnosis [43]. This, in turn, is due to several reasons, and one of these is the lack of specific and suitable screening tools. The available screening approaches are able to reduce mortality, but there is the possibility of false positive results, and some tests are invasive and not devoid of risk of harm for patients [44]. Differently from what happens for BCa, the lack of preventive suitable tools together with the absence of clear and specific symptoms make the early diagnosis of OvCa very difficult. If we compare OvCa to all the types of cancers, it represents the 11th malignancy for incidence and the 5th for mortality among the female population. It has been recently considered that the inclusion of conventional drugs in nanomaterials for their more specific and efficient delivery could also represent a fruitful therapeutic approach for this kind of cancer [17].

Even though multiple preclinical and human observational studies investigated the potential use of omega-3 PUFAs as adjuvants against OvCa [39,45], alone or in combination with conventional therapy, interventional clinical trials are still lacking, and their inclusion in nanoformulations has not been so far investigated. Thus, we will briefly focus on the results obtained in the most recent years on the effects of omega-3 PUFAs against OvCa to understand if, altogether, they may represent the needed basis to move to interventional clinical trials and the development of omega-3 PUFAs containing drug-delivery nanosystems for ovarian cancer. Finally, we will discuss future perspectives for the study of omega-3 PUFA-based nanoformulations against OvCa.

## 2. Omega-3 PUFA-Based Nanoformulations against BCa

We recently demonstrated that the inclusion of omega-3 PUFAs in nanomaterials (xanthan gum-based microspheres and hydrogels) improved their cancer activities against colon cancer cells [14]. Moreover, in a review of the literature recently published by us [15], we analyzed the results of reports published until July 2018 and focused on the potential advantages of using omega-3 PUFA-based nanomaterials for the therapy of cardiovascular diseases and some of the most prevalent cancers, including BCa. However, this is a field of growing interest, and, since then, other interesting results have been obtained in the BCa research field, which will be analyzed in detail here.

The possible antineoplastic use of omega-3 PUFAs included in nanoformulations is still attracting great attention as demonstrated by the promising results obtained over the last few years in a large array of cancers, such as hepatocarcinoma [46], neuroblastoma [47], pancreatic cancer [48] and melanoma [49] (for a review, see [50]).

One of the most investigated BCa therapeutic nano-based approaches using combined treatments of omega-3 PUFAs and conventional drugs is the inclusion of both DHA and doxorubicin (DOX) in nanostructures. DOX and other anthracyclines, such as epirubicin, are largely used for the adjuvant therapy of BCa, and have a similar response rate. However, they may cause dramatic side effects in normal tissue, such as cardiotoxicity, nephrotoxicity and myelosuppression [51]. On the other hand, among the omega-3 PUFAs, DHA is the most used for potentiating the anti-cancer effect of DOX, since it is considered the most effective. This effect has been related to its six double bonds in the carbon chain, which makes this FA highly peroxidable, and able to amplify the oxidative-dependent cytotoxicity of anthracyclines towards tumor cells [52]. Cancer cells show both an increased ROS production and a decreased ROS-scavenging capacity, which is known as an oxidative stress condition and is believed to play a role in the initiation and progression of cancer [53]. Moreover, cancer cells, among other abnormal features, exhibit a reduced ability to withstand possible damage induced by increased intracellular ROS levels induced by exogenous pro-oxidant agents [52].

There are solid experimental bases for the DOX/DHA nano-combinatory strategy, since many findings obtained during the last two decades in experimental studies conducted both in vitro and in vivo, as well as in BCa patients, have substantiated the hypothesis that dietary omega-3 PUFAs are able to potentiate the antineoplastic effects of anthracyclines [37,39,54,55,56,57] as well as to reduce the systemic side effects induced by them. In particular, several human interventional studies have been so far performed (for a review, see [39]) to study the antineoplastic effects that omega-3 PUFAs may exert against BCa in combination with anthracycline-based chemotherapy. The first one was the early human phase II trial of Bougnoux et al. [58], which demonstrated that a DHA supplementation, and its consequent increased incorporation in plasma, were directly associated with more extended time to progression and overall survival in BCa patients treated with anthracyclines. Then, in the last few years, interesting developments have been made in the field and are reported in Table 1. Following the first demonstration of the antineoplastic effect of lipid nanostructures carrier (NLC) co-loaded with DOX and DHA in spheroid BCa models [59] published several years ago; more recently, the same authors [19] radiolabeled at the surface of the same NLC-DOX-DHA formulation with a radioactive tracer (Tc-99m) to obtain a nanotheranostic platform combining and delivering both a therapeutic (DOX + DHA) and a diagnostic agent for a bioimaging approach. In particular, TC-99m was chosen for its physical-chemical properties, availability and low cost, making it a widely used agent for nuclear imaging tests, such as single-photon emission computerized tomography [60]. The nanoformulation, prepared by the hot melting homogenization method as described in the first report [59], has a lipid matrix composed of compritol and DHA in triglyceride form. The new nanoformulation, however, was not constructed as in the first preparation by encapsulating the Tc-99m-DOX complex in the SLN by ion-pair with oleic acid, since the complexation with technetium competes for the same DOX amino group used for its ion-pairing with oleic acid. Thus, being DOX a hydrophilic drug, it was not incorporated into the hydrophobic NLC core and remained in a superficial position bound to the radiotracer. The newly designed 99mTc-radiolabeled NLC-DHA-DOX formulation had a small size (70 nm) and a prolonged circulation time. The authors used these nanoparticles for scintigraphic images and biodistribution studies in BALB/c mice injected with 4T1 breast cancer cells. The images showed an increased tumor uptake with respect to the controlateral muscle and a higher [99mTc]-NLC-DHA-DOX accumulation in the tumors with respect to [99mTc]-DOX alone. Consequently, a more efficient tumor growth inhibition was observed after the treatment with [99mTc]-NLC-DHA-DOX compared to the free DOX or [99mTc]-NLC-DOX treatments.

Instead, Kim et al. [61] constructed glycol chitosan (GC)-DHA liposomes for the delivery of both DOX and rapamycin (RAPA) and evaluated them in drug-resistant BCa cells (MDA-MB-231). RAPA is a specific inhibitor of mTOR survival signaling, whose activation has been involved in drug resistance, and GC is a safe polysaccharide polymer positively charged showing mucoadhesive properties and possessing easily modifiable chemical functional groups [62]. The rationale for using both GC and DHA as carriers was related to the fact that the drugs to be delivered (DOX and RAPA) had totally different physicochemical properties, and it was challenging to enclose both of them in one single particle. Thus, the authors used the strategy of preparing DHA liposomes containing RAPA and then mixing them with the DOX-conjugated-GC to obtain the GC-DOX/RAPA-DHA liposomes. This GC-DHA liposomal formulation allowed a sustained release of the loaded drugs, especially in acidic conditions, typical of tumors. In addition, it allowed to co-deliver directly to the tumor cells an additional antineoplastic factor such as DHA, known to be beneficial in cancer therapy. However, the authors evaluated only the cytotoxicity of RAPA-DHA liposomes or GC-DOX conjugates as compared to that exhibited by the GC-DOX/RAPA-DHA formulation, leaving no room for understanding the possible specific role exerted by DHA in potentiating the inhibition of cancer cell growth.

Tripathi et al. [63], instead, used a DOX-loaded folate functionalized nanoemulsion (f-DOX-NE) containing ALA in the lipid phase for the targeted delivery of DOX to MCF-7 cell lines in vitro or to 7,12-dimethylbenz[a]anthracene-induced mammary gland carcinoma in female Albino Wistar rats. In the in vitro conditions, they found that the new formulation enhanced the DOX-related cell growth inhibition through the induction of ROS production and by enhancing the mitochondrial pathway of apoptosis. These results agreed with what they observed in vivo, where the treatment with f-DOX-NE reduced significantly, and more than DOX given alone, the tumor volumes, and induced the downregulation of anti-apoptotic factors (Bcl-2 and MMP-9) as well as the up-regulation of pro-apoptotic (caspase-9 and Bax) factors. Moreover, the authors found an increased animal survival and decreased cardiotoxicity, and related all these effects to the protective action of ALA. The inclusion of ALA in nanoformulations is interesting, particularly in the case of HER2 overexpressing BCa, since it was previously shown [64] that dietary treatment with flaxseed oil (where ALA medium content is about 50% of total FAs [65]) was able to inhibit the abnormal HER2 signal transduction pathway of MCF-7 tumors [66] growing in nude mice and, through this mechanism, to induce apoptosis. The inhibition of HER-2-related MAPK pathways and the induction of apoptosis were also in agreement with what was previously observed by Ciocci et al. [67] in MCF-7 cells treated with nanoemulsions of ALA and diallyl disulfide.

A recent report [68] also demonstrated that nanoparticles containing ALA conjugated with paclitaxel (ALA-PTX NPs) increased BCa cell uptake and anticancer activity both in vitro and in vivo. As underlined by these authors [68], the use of ALA may be useful since small hydrophobic molecules (such as ALA) enclosed in a nanostructure allow higher drug loading, simple synthesis, and better biocompatibility compared to heavier and more complex molecules with comparable hydrophobicity features. However, it should be understood that only a few studies have so far preferred ALA to DHA or EPA for inclusion in nanotherapeutic formulations. The very long-chain PUFAs EPA and DHA have generally shown higher protective (anti-inflammatory and antineoplastic) effects as compared to ALA [69,70], and the beneficial effects of ALA are usually related to its metabolic conversion into EPA and DHA. In our opinion, on this basis and from a translational point of view, there is an advantage in directly enclosing DHA or EPA in the nanoformulations, and it is related to the observation that the efficiency in performing the elongation and desaturation steps needed to obtain EPA and DHA from ALA is not equally distributed among the different individuals of the human species, and along the different ages [29,71,72].

In their recent report, Li et al. [73] also showed an enhanced anticancer effect of PTX on MCF-7 cells cultured in vitro or injected in nude mice when this drug was administered combined with DHA in a lipid nanoemulsion (LN) decorated with folic acid (FA) in order to improve the delivery to cancer cells. The nanoemulsion was obtained by mixing DHA and PTX with cholesterol, egg phosphatidylcholine and DSPE-PEG2000-FA. The results obtained in vitro demonstrated that DHA enhanced the cytotoxicity of PTX to BCa cells both when PTX and DHA were added in combination as free compounds, and especially when they were co-encapsulated in LNs. In vivo, the inclusion of DHA in the PTX-LN resulted in significantly decreased volumes of the growing tumors, the increased survival rate of tumor-bearing mice and decreased weight loss; thus, demonstrating the efficacy of the PTX-DHA combination. Moreover, the LN with FA decoration on their surface allowed us to obtain even better results, since tumor cells show a high density of FA receptors at their surface that drives and enhances LN uptake [74]. The presence of either DHA or FA in the LN also increased the uptake of PTX by M2 macrophages, and the maximum effect was observed when both compounds were enclosed in the nanoformulation. That resulted in an increased number of M1 in comparison to M2 macrophages, which led to a tumor microenvironment favoring the immune control of tumor growth, and delayed tumor progression [75].

Other recent reports further investigated nanoformulations containing the combination of DOX and omega-3 PUFAs, with the main aim of improving the delivery of DOX in BCa [76,77]. In one of the reports [76], DOX was included in a nanosystem with mixed PUFAs (MPUFAs-DOX@liposomes) containing high levels of omega-3 PUFAs. When the authors evaluated this newly designed nanoformulation on MCF-7 BCa cells, MPUFAs-DOX@liposomes showed higher toxicity than free DOX or DOX@liposomes. The authors ascribed the enhanced antineoplastic effect to several factors, such as: (i) the increased lipophilicity of the nanoformulation, which improved the drug loading efficiency; (ii) the faster DOX release rate from the omega-3 PUFA nanosystem at pH 5.0 than at pH 7.4, that may potentially decrease the secondary toxic effects of this drug; and, finally, (iii) the improved uptake of DOX when delivered to the neoplastic cells through the hydrophobic-lipid rich nanosystem. However, they did not consider the recognized and more specific antineoplastic activities of these FAs, as well as the increased efficiency shown by DOX as it was combined with a concomitant treatment with omega-3 PUFAs, as previously reported by other authors [37,46,61].

Instead, there are two recent reports by Lages et al. [77,78] where the omega-3 PUFA DHA was chosen for its potential to increase the delivery of DOX and α-tocopherol succinate (TS), but also for its specific anticancer properties. TS was also previously shown to have direct antineoplastic activities and to act in synergism with chemotherapeutic drugs [79]. These authors evaluated their nanoformulations on 4T1 murine BCa cells, either cultured in vitro or implanted in syngeneic Balb/C mice. In the first report [78], they obtained elevated levels of DOX encapsulation (almost 100%) by ion-pairing with TS. DOX-TS had increased lipophilicity as compared to DOX, and that facilitated the encapsulation in the DHA lipid matrix. In the in vitro experiments, they observed antitumoral synergistic effects of the three components of the nanoformulation (DOX, DHA and TS) and increased cellular uptake of DOX with respect to the free DOX. These authors confirmed the improved efficiency in their in vivo studies, where they found that the nanoformulation was able to induce a significant inhibition of 4T1 BCa cell growth, and their ability to form metastases. Moreover, it prolonged substantially the survival of the implanted mice and reduced the DOX-induced liver- and cardio-toxicity. However, in their second study [77], the same authors judged unsatisfactorily the pharmacokinetic characteristics of their initially synthesized formulation [78], and thus, tried to improve its efficiency by synthesizing a covalent DOX-TS conjugate (by hydrazone bond) that they included in the DHA-based nanocarrier. When the newly developed nanoformulation was assessed in vitro, a slow DOX release was observed at pH 7.4, but a fast release was observed under acidic conditions, due to the pH-sensitive hydrazone bond. Considering the low-tumor-cell pH conditions, this represents a high-valuable feature for a drug that should be precisely delivered to the tumor area. The in vivo study confirmed that the co-encapsulation in the nanoformulation of a powerful antioxidant such as TS and an antineoplastic agent such as DHA, could decrease the DOX-induced heart and liver damage, and synergize with DOX in reducing tumor growth.

DHA was reported to induce the expression of BCa non-coding microRNA *let-7a*, which functions as a tumor suppressor and, when it is reduced, is associated with tumorigenesis, poor prognosis and chemoresistance in different cancers, including BCa [80,81,82]. Since the chemoterapic drug 5-fluorouracil (5-FU) had been observed to exert the same inducing effect as DHA on *let-7a* expression [83,84], Li et al. [85] designed a nanosystem with magnetization for the delivery of 5-FU consisting of a porous zirconium-based metal-organic framework (MOF) UIO-66-NH2 bound to DHA. The authors chose this MOF for its feature of low cytotoxicity and high biocompatibility, which makes it an optimal carrier for drug delivery [86]. Interestingly, from our point of view, they found that when the delivery system contained DHA, the inducing effect of 5-FU-MOF on BCa cell apoptosis observed by them in vitro was enhanced, whereas the BCa cell viability was further reduced, and the level of *let-7a* further increased. Moreover, they observed that the presence of DHA in the nanoformulation enhanced the suppressive activity of the 5-FU-MOF system on the tumorigenic activity of BCa cells in nude mice in vivo *and* acted in synergism with the 5-FU-MOF in enhancing the chemosensitivity of BCa cells. Most interestingly, the authors suggested that if the sensitivity of BCa cells to 5-FU was increased by the presence of DHA, it was due to the FA ability to further induce the up-regulation of *let-7a* expression.

**Table 1 ijms-23-12032-t001:** Omega-3 PUFA containing nanosystems evaluated for BCa therapy (2018–2022).

Preclinical BCa Model	Nanoformulation	Antineoplastic Drug/Nutraceutics Associated to ω-3 PUFA in the Nanoformulation	Omega-3 PUFA in the Nanoformulation	Anticancer Effect(s) of Nanoformulation Respect to the Drug Alone	Protection Induced by the Nanoformulations vs. Free Drug-Induced Side-Effects In Vivo	Ref.
4T1 tumor-bearing BALB/c mice	Nanoemulsions radiolabeled with Tc-99m (for theranostic use)	DOX radiolabeled with TC-99m	DHA	↓ Tumor volume	↓ Body loss ↓ hepato- and cardio-toxicity	[19]
In vitro: DOX-resistant MDA-MB-231 cells	Liposomes (DHA liposomes containing RAPA + DOX-conjugated glycol chitosan)	DOX -RAPA	DHA	↓ IC50 ↓ Cell proliferation	ND	[61]
In vitro: MCF-7 cells In vivo: DMBA-induced BCa	Folate functionalized nanoemulsions	DOX	ALA	↓ MCF-7 cell viability ↑ MCF-7 apoptosis ↓ DMBA-induced BCa volume ↑ tumor targeting	↓ Body loss ↓ Cardiotoxicity ↑ Animal survival	[63]
In vitro: MCF-7 cells, MCF-7/ADR cell In vivo: MCF-7/ADR Cell xenografted in BALB/c nude Mice	Nanoparticles (ALA-PTX conjugate) formed by a precipitation method	PTX	ALA	↓ Tumor volume	↑ Mouse survival	[68]
MMCF-7 cells MCF7 sublines resistant to cisplatin (MCF-7/DDP) MCF-7 cells in nude mice	Lipid nanoemulsions decorated with folic acid	PTX	DHA	↓ In vitro: Cell viability ↑ Apoptosis ↑ Uptake by MCF-7 cells and M2 macrophages In vivo: ↓ Tumor volume growth	↓ Heart and lung toxicity ↑ Mouse survival time	[73]
In vitro: MCF-7 cells	MPUFAs-DOX liposomes	DOX	Mixed PUFAs with high omega-3 PUFA content	In vitro: ↑ Cytotoxicity	ND	[76]
In vitro: 4T1 murin breast cells In vivo: 4T1 implanted in BALB/c mice	Nanoemulsions	DOX + α-Tocopherol succinate (TS) DOX-TS covalent conjugate [79] DOX-TS ion-pair [80]	DHA	↓ Tumor growth rate	↓ Suppression of DOX- induced mortality in tumor-bearing mice Hepatotoxicity (↓ AST levels) ↓ Cardiotoxicity	[77,78]
In vitro: MCF-7 cells In vivo: DMBA-induced BCa	A drug delivery system with magnetization [porous zirconium-based metal-organic framework (MOF) UIO-66-NH2 bound to DHA]	5-FU	DHA	In vitro: ↓ Cell viability ↑ Apoptosis In vivo: ↓ Tumor volume and weight ↑ BCa cell chemosensitivity	vs. Control mice *: Unaltered serum biochemical parameters (ALP, ALT and AST); Unaltered complete blood count	[85]

Tc-99m: metastable nuclear isomer of technetium-99; ALA: α-linolenic acid; 5-FU: 5-fluorouracil; DHA: docosahexaenoic acid; DMBA: dimethyl benzanthracene; DOX: doxorubicin; EPA eicosapentaenoic acid; PTX: paclitaxel; 5-FU: 5-fluorouracil; MPUFAs-DOX: DOX) and mixed polyunsaturated fatty acid (MPUFA)-DOX ion-pairing; * In this case the comparison was made only between animals treated or not (control animals) with the DHA containing nanosystem.

## 3. Omega-3 PUFAs and OvCa: What Is Known from Preclinical In Vivo Studies

Over the last two decades, a consistent body of literature has been accumulated demonstrating, both in vitro and in vivo, the potential antineoplastic efficacy of omega-3 PUFAs by using preclinical models of OvCa (see below for details). On the other hand, there are limited numbers of human observational studies, especially if compared to those available for BCa, and often their outcomes are also inconsistent [71,87]. Moreover, there are no clinical interventional trials to date [39]. This may have discouraged researchers from developing omega-3 PUFA-containing nanoformulations to improve OvCa therapy. For these reasons, it may be important to have a picture of the current knowledge in this field of research to better orientate future developments of the research both in the clinical setting and in the nanotherapy area. To this aim, we will describe and critically analyze the most relevant findings obtained with omega-3 PUFA treatments in preclinical in vivo models of OvCa, which are better than the in vitro ones that may mimic the human condition (Figure 1).

Recently, West et al. [88] reported the ability of DHA to inhibit the growth and proliferation of OvCa developing in a transgenic mouse model of OvCa in vivo (the KpB transgenic mouse model of high-grade serous epithelial OvCa mouse model (K18-gT121 ±; p53fl/fl; Brca1fl/fl; KpB)). In this work, DHA was intraperitoneally (IP) injected at the daily dose of 15 mg/kg for 28 days. The intraperitoneal way of administration appears very appropriate since the most common way of OvCa spreading is through the direct exfoliation of cancer cells into the peritoneal cavity [89]. It was also shown that the drug intraperitoneal (IP) injection increases their half-life and concentrations in the target of OvCa cells [90]. Nanoparticles encapsulating another nutraceutical, the epigallocatechin gallate, were recently administered IP in a peritoneal metastatic murine model of human OvCa [91]. The recent findings of West et al. [88] are extremely interesting, since their work is the first preclinical work where the anticancer effects of an omega-3 PUFA were investigated in an immunocompetent mammalian OvCa model, especially considering the crucial role exerted by the tumor microenvironment immune cells in the induction of the growth and progression of tumors. Instead, another early work was performed using SCID mice transplanted with paclitaxel-sensitive human ovarian carcinoma A121 cells [92]. That report also demonstrated that DHA covalently conjugated to the chemotherapeutic second-generation taxoid paclitaxel was able to potentiate the anticancer activity of this drug, inducing the complete regression of the A121 cell xenografts. Interestingly, when these conjugates were used by the same authors [92] against drug-resistant human colon cancer cells implanted in SCID mice, they were able to overcome the resistance and, also in this case, differently from paclitaxel alone and paclitaxel-DHA, induced the total regression of the tumor xenografts. A similar report was recently published by Udumula et al. [93], who implanted the human OvCa SKOV3ip or CaOV3 cells, in the peritoneal cavity of nude mice and then dietary supplemented the mice with 100 mg/kg EPA or DHA administered by gavage per 4–5 weeks. The antineoplastic efficacy of the two FAs was different, and dependent on the cell line implanted. In the SKOV3ip xenograft model, both the FAs resulted as less active than in the CaOV3 one. EPA resulted more efficacious than DHA, and increased the median mouse survival of control mice (23.5 days, corn oil-treated mice) by 97.8%, whereas DHA induced only a 23.4% increase. Moreover, just EPA reduced markedly and significantly the size of the excised tumors, as well as the Ki-67 proliferation index of the OvCa cells grown in vivo, as compared to the cells grown in control mice.

Instead, another work evaluated the ALA antineoplastic activity. In this case, another animal model was used, the xenograft model of zebrafish [94], which is farther than the xenograft mouse models in terms of evolutionary distance and phylogenetic trees from the human species. Anyway, this model was recently considered a helpful model for the study of metastasis in vivo [95], since it has the advantage of mirroring the degree of invasion/metastasis in vitro and in mice, and it is able to furnish information in a faster and less expensive way compared to mouse models of metastasis. In fact, in the zebrafish model, the human cancer cell lines are injected into the perivitelline space of two-days old larvae, where they can survive and metastasize, since the larvae will not develop an adaptive immune system until 14 days post-fertilization [96]. In Wang et al.’s work [94], performed on the zebrafish larvae (at 1-day post-fertilization) only DHA was able to suppress the development of cell metastasis. From the in vitro experiments, the authors suggested that this effect could be related to the DHA ability to suppress the NF-κB signaling.

**Figure 1 ijms-23-12032-f001:**
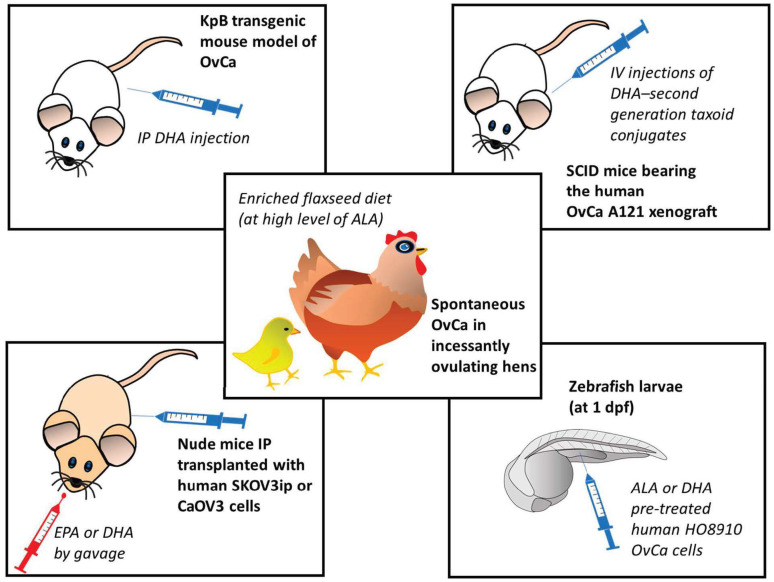
Preclinical in vivo models of ovarian cancer used in studies for the evaluation of the anticancer effects of omega-3 PUFAs [88,92,93,94,97]. ALA: α-linolenic acid; DHA: docosahexaenoic acid; dpf: day after fecundation; EPA: eicosapentaenoic acid; IP: intraperitoneal; IV; intravascular; OvCa: ovarian cancer. The numbers in square brackets indicate the references in which the respective animal models were used.

Over the last two decades, a series of in vivo studies had been also conducted using a model of spontaneous OvCa frequent in incessantly ovulating hens (for a review see: [97]). This model, even though again not using mammalians, has the advantage of being the only known animal model of spontaneous OvCa, and it has been so far considered useful for pre-clinical and dietary intervention studies focused on OvCa prevention, since the OvCa developed in the hen is histologically similar and shares similar features with the human OvCa considering the ascites and metastasis formation in peritoneum [98,99]. The research developed over the years by using this model has demonstrated that a flaxseed-rich diet can decrease the severity and incidence of OvCa [99]. Moreover, it was observed that this diet promotes apoptosis and suppresses angiogenesis in the OvCa, but not in the normal avian ovaries [99]. Interestingly, besides containing high levels of ALA, which can then be metabolically converted into the long-chain EPA and DHA, flaxseed is also a source of the phytoestrogen lignan secoisolaricisresinol diglucoside, which is converted to enterodiol and enterolactone, both having anti-estrogenic and antioxidant properties [99]. Moreover, the enriched flaxseed diet was shown to induce the CYP1A1 pathway, and through it, the formation of 2-hydroxy estradiol metabolites, which are then converted into 2-methoxyestradiol (2MeOE_2_) estrogen, which is the least powerful estrogenic metabolite as compared to the others (such as 4-hydroxy and 16-hydroxy metabolites) produced by the different CYP enzymes (CYP3A4 and CYP1B1, respectively) [100]. This will result in a higher 2-hydroxy/16-hydroxy estradiol ratio, previously shown to be protective against postmenopausal BCa [101]. On these bases, Pal et al. [99] recently also investigated the effects of individual treatments with DHA or 2MeOE_2_ in vitro in human OvCa cells, and in an in vivo model that mimics the neo-angiogenetic process in vivo, i.e., the chorioallantoic membrane assay [102]. Using the two models, the authors demonstrated that, whereas only 2MeOE_2_ induced apoptosis, both the compounds acted by inhibiting neo-angiogenesis, even though with different molecular mechanisms. These results suggest a possible 2MeOE_2_/DHA combinational approach for the treatment of advanced OvCa. Therefore, the inclusion of both compounds in newly developed nanoformulations to be evaluated for their potential to contrast OvCa would deserve particular attention. Overall, the in vivo studies substantiate the hypothesis that omega-3 PUFAs may represent potential preventive/therapeutic agents against not only BCa, but also OvCa. Moreover, they suggest that their anticancer potential, when enclosed in nanoformulations in combination with other nutraceuticals or chemotherapeutic drugs, would be worth to be evaluated.

A considerable number of in vitro studies were also performed to evaluate the effects of omega-3 PUFAs on OvCa cells, but we will not analyze them here, both for the sake of brevity and being this issue beyond the scope of this review. However, it is interesting to underline that in a comparable number of these studies the cells were treated either with EPA or DHA, and just in two of them [93,103] both the two long-chain PUFAs were separately evaluated to make a comparison between them. In one case, DHA showed more pronounced anticancer properties [103], whereas in the other one EPA resulted in being the most active [93]. Therefore, robust indications are not yet available for choosing one or the other PUFA for further OvCa in vitro studies. In this context, also the hypothesis put forward very recently by Udumula et al. is of interest [93]. Based on their in vitro findings, these authors suggested that metformin, a “repurposed” drug for OvCa, previously reported to have anticancer effects in preclinical and human observational studies [104], could exert its powerful antineoplastic effects in OvCa by enhancing the endogenous production of EPA and DHA from the precursor ALA [93]. This stimulating hypothesis could represent the rationale for the evaluation of new treatments for OvCa both in vitro and in vivo, where the anticancer properties of metformin could be enhanced by the combined treatment with omega-3 PUFAs, furnishing all these antineoplastic agents (metformin and EPA or DHA) either in their free forms or co-delivered in newly developed nanoformulations. In our opinion, the reason why studies focused on the inclusion of omega-3 PUFAs in nanoformulations designed for the therapy of OvCa have not yet been performed, even despite all the robust and consistent data obtained in the preclinical setting, may be especially related to the inconsistency of the outcomes of the human observational studies performed in this field [71,87].

## 4. Conclusions

Multiple studies have so far investigated the effects of the omega-3 PUFA inclusion in nanoformulations designed for drug delivery in BCa. Our critical analysis of the most recent results of these studies indicated that these FAs were able to potentiate the antineoplastic effects of the co-enclosed drugs (DOX, 5-FU or PXT), as well as to enhance the drug delivery to target sites, and minimize the drug-induced side effects. However, the mechanisms at the basis of the omega-3 PUFA effects are often not investigated in detail in the articles analyzed. In our opinion, in the future, it would be very valuable to investigate the nanoformulations also from a mechanistic point of view.

Conversely, evaluations of the potential effects of omega-3 PUFAs enclosed in nanoformulation designed for the therapy of OvCa are not yet available. This appears quite odd, since the anti-neoplastic efficacy of these fatty acids has been convincingly demonstrated by a series of studies using preclinical in vivo models of OvCa. We have critically analyzed here the results of these studies and concluded that they seem to provide sufficient evidence for: (a) designing for years to come still lacking interventional clinical trials investigating the possible adjuvant effects that omega-3 PUFAs may elicit in combination with antineoplastic drugs; (b) planning in the next future studies aimed to evaluate the possible advantage related to the inclusion of omega-3 PUFAs in drug-delivery nanosystems designed for OvCa.

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
