# Peer review of "Nutraceutical-Based Nanoformulations for Breast and Ovarian Cancer Treatment"

_ijms, 2022, doi:10.3390/ijms231912032_

Round 1
Reviewer 1 Report
Overall the manuscript is well written and presented. However some recommended corrections and clarifications are required to be incorporated in the revised manuscript.
Check the manuscript for grammatical and typo errors. Check all the references for correctness and journal referencing style. Add some recent references in the introductionAuthor Response
Overall, the manuscript is well written and presented.
We thank the Reviewer #1 for the appreciation of our work.
However, some recommended corrections and clarifications are required to be incorporated in the revised manuscript. Check the manuscript for grammatical and typo errors.
We have now checked throughout the manuscript and tried to eliminate all the grammatical and typo errors.
Check all the references for correctness and journal referencing style. Add some recent references in the introduction.
We have also checked the references for correctness and changed them when needed, following the journal reference style. We have eliminated for incorrectness the old ref. 17, as well as some old references in the Introduction section (the old ref.: 1, 7, 29, 30,35 and 37), and have added more recent references, as requested (new references 7, 18,29,30,42). However, we have left the refs. 8-11 (dating 2004-2005-2009 and 2014), since in the relative sentence [lines 34-38], we were specifically referring to our past work focused on omega-3 PUFAs combined with other natural compounds. In fact, to better clarifying this point we have now indicated at the beginning of the sentence “ In the past, we explored whether combination…”

Reviewer 2 Report
The review entitled "Nutraceutical-based nanoformulations for breast and ovarian cancer treatment " seems report not review article. It requires more figure figures to explain the possible mechanistic approach . Mechanisms of nanoformulations in cancer treatment can discussed with recent work of the field. https://doi.org/10.2147/IJN.S285392, https://doi.org/10.1021/acsomega.1c01467
Conclusion can be revised by adding future directions
Author Response
The review entitled "Nutraceutical-based nanoformulations for breast and ovarian cancer treatment " seems report not review article. It requires more figure figures to explain the possible mechanistic approach . Mechanisms of nanoformulations in cancer treatment can discussed with recent work of thefield. https://doi.org/10.2147/IJN.S285392, https://doi.org/10.1021/acsomega.1c01467
In the now enclosed two-page Table 1 entitled: Omega-3 PUFA containing nanosystems evaluated for BCa therapy (2018-2022) plenty of information on nanoformulation can be found, including mechanistic information related to increased apoptosis, increased cytotoxicity, increased cancer cell chemosensitivity, etc.
We thank the Reviewer #2 for the indication of two articles by Maqusood Ahamed’s group, where the characterization of nanoparticles (not containing nutraceuticals and not evaluated in BCa or OvCa) was performed and where particular attention was dedicated to the identification of the main mechanism (oxidative) involved in the nanoparticle anti-cancer effects.
In agreement, in our review, we had discussed how the oxidative mechanism may be involved in the antineoplastic activity of DHA in BCa, especially when it was combined with other drugs with pro-oxidant effects (such as Doxorubicin, DOX, and other anthracyclines) (see in the paragraph 2. Omega-3 PUFA-based nanoformulations against BCa 2, lines 183-192). Mechanistic information was reported also at lines 250-256 [oxidative potential and proapoptotic potential of DOX increased in folate functionalized nanoemulsion containing alpha-linolenic acid (ALA, see lines 247-257)]. Other mechanisms were discussed, for instance for the same nanoformulation, it was discussed the inhibitory effect of ALA on HER2 expression (lines 258-265). Other discussed mechanisms were the increased hydrophobicity of the nanoparticles containing PTX and DHA for the presence of DHA or LNA (line 266-271) or MPUFAs-DOX@liposomes (line 306-311) that could favor their uptake by cancer cells. Moreover, it was interesting the demonstration of an easier uptake of the nanoparticles containing PTX and DHA by cancer cells as well as by M1 macrophages which could have led to a tumor microenvironment favoring the immune control of tumor growth, and a delayed tumor progression (lines 294-299).
However, not all the articles concerning nanoformulations based on Omega-3 PUFA in combination with other drugs and to be used for BCa explored the mechanisms potentially involved in their antineoplastic effects. We have now underlined this limitation in the Conclusions and the need for the future to investigate the possible mechanisms involved (lines 491-495).
Conclusion can be revised by adding future directions
We have now added in the Conclusions section the future directions for possible studies in the field of BCa, omega-3 and nanoformulations. Moreover, we have now written in a more comprehensive way the future directions for possible OvCa studies in this field.
